# OPTIMAL TRANSPORT-BASED SUPERVISED GRAPH SUMMARIZATION

## ABSTRACT

Graph summarization is the problem of producing smaller graph representations of an input graph dataset, in such a way that the smaller *compressed* graphs capture relevant structural information for downstream tasks. One graph summarization method, recently proposed in Garg & Jaakkola (2019), formulates an optimal transport-based framework that allows prior information about node, edge, and attribute importance to be incorporated into the graph summarization process. We consider the problem of graph summarization in a supervised setting, wherein we seek to preserve relevant information about a class label. We first formulate this problem in terms of maximizing the Shannon mutual information between the summarized graph and the class label. We propose a method that incorporates mutual information estimates between random variables associated with sample graphs and class labels into the optimal transport compression framework. We empirically show performance improvements over previous works in terms of classification accuracy and time on synthetic and certain real datasets. We also theoretically explore the limitations of the optimal transport approach for the supervised summarization problem and we show that it fails to satisfy a certain desirable information monotonicity property.

## 1 INTRODUCTION

Machine learning involving graphs has a wide range of applications in artificial intelligence Scarselli et al. (2008); Dessì et al. (2020), network analysis, and biological interactions Han et al. (2019); Chen et al. (2020). Graph classification problems use the network structure of the underlying data to improve predictive decision outcomes. However, graph datasets are often enormous, and the algorithms used to extract relevant information from graphs are frequently computationally expensive. Graph summarization addresses these scalability issues by computing reduced representations of graph datasets while retaining relevant information. As with numerous other problems in machine learning, the precise meaning of "reduced representation" does not have one single mathematical definition, and there is no single objective function being optimized. There are thus various approaches to this problem. For a survey, see Liu et al. (2018). The particular type of approach of interest in this paper takes a dataset of graphs and a number $k$ as input and outputs, for each graph $G$ in the dataset, a subgraph $H \subseteq G$ is induced by $k$ vertices.

Optimal transport, the general problem of moving one distribution of mass to another as efficiently as possible, has been used in many recent graph-related problems, such as graph matching via the Gromov-Wasserstein distance Xu et al. (2019). One recent approach to the graph summarization problem that allows for the incorporation of user-engineered prior information is the Optimal Transport based Compression (OTC) approach of Garg & Jaakkola (2019). Their approach is as follows: a graph $G$, a target number $k$ of vertices, a probability distribution $\rho_0$ on the vertices of $G$, and a cost function $c : E(G) \to \mathbb{R}$, where $E(G)$ denotes the set of edges of $G$, are given as input. A probability distribution $\rho_1$ is computed by minimizing the Wasserstein distance on $G$ between $\rho_0$ and $\rho_1$ with respect to the cost function $c$, with the constraint, that the number of vertices in the support of $\rho_1$ is at most $k$. The output subgraph $H$ is the one induced by the vertices in the support of $\rho_1$. Prior information can be incorporated into the method via appropriately choosing $\rho_0$ and $c$, but in the prior work, this "prior information" is not learned, and $\rho_0$ and $c$ are set heuristically.

In the present work, we propose a novel supervised summarization algorithm based on Optimal Transport that estimates principled values for the parameters from input data. We show that it empirically surpasses the state-of-the-art performance (including the performance of the specific method proposed in OTC) on selected real and synthetic datasets. The novelty of the summarization algorithm is that we set our optimal transport parameters in terms of node attributes' and edge indicators' information about class variables. Along the way, we develop an estimator for the mutual information between a latent position vector graph and a class label. This extends the well-studied problem of information-theoretic measure estimation Kraskov et al. (2004); Moon et al. (2017); Noshad et al. (2017). Our estimator is inspired by EDGE's fast and relatively accurate implementation Noshad et al. (2019).

Toward providing a theory for the limitations of our approach to graph summarization, we propose a natural information-theoretic Cover & Thomas (1991); Yasaei Sekeh et al. (2018) objective function for the task of supervised graph summarization and show that it is NP-hard to optimize. Using this new framework, we explore the limitations of the optimal-transport-based approach, both theoretically and empirically. Specifically, we formulate a notion of information monotonicity of an optimal transport parameter pair with respect to a data distribution. This is the desirable property that means the flow cost decreases monotonically as the mutual information of the resulting summarized graph data with the corresponding class labels increases. This means that optimizing flow cost increases class label information (but may not optimize it). We show that any optimal transport parameter pair satisfying natural properties fails to exhibit information monotonicity for at least some data distributions.

**Contribution:** We summarize our contribution in this paper as follows: We *propose* a novel information-theoretic attributed graph summarization problem formulation in which the goal is to choose a graph summary that maximizes the mutual information between sample graphs and class labels. We *theoretically* prove that the problem of maximization of the mutual information between attributed graphs and class variables with the knowledge of the data distribution is NP-hard, even approximately. We then show via explicit constructions some limitations of the optimal transport graph compression approach in Garg & Jaakkola (2019)(OTC). We then *introduce* a supervised graph summarization framework based on optimal transport and experimentally show that it outperforms the baseline (no compression) and the unsupervised OTC method in terms of post-compression classification time and test accuracy, despite suffering from the fundamental limitations of our theoretical contribution.

## 2 PROBLEM FORMULATION: SUPERVISED GRAPH SUMMARIZATION

We formulate the supervised graph summarization problem as follows. We fix a target compression ratio $\kappa \in (0, 1)$, which will be the ratio of the number of vertices in a summarized graph to the number in the original graph. A probability distribution $\mathcal{D}$ over tuples $(G, X, C)$ is fixed by nature and unknown to the summarizer and the classifier. Here, the graphs $G$ are defined on a single common set $V$ of vertices, $X$ is a $|V| \times d$ matrix whose rows are $d$-dimensional feature vectors corresponding to the vertices of $G$, and $C$ is a class label coming from some fixed set $\mathcal{C}$. A dataset $\{(G_i, X_i, C_i)\}_{i=1}^m \sim \mathcal{D}^m$ consisting of $m$ independent and identically distributed (iid) samples from $\mathcal{D}$ is presented to us. Our task is, given sample $\{(G_i, X_i, C_i)\}_{i=1}^m$ but no knowledge of distribution $\mathcal{D}$, to select a subset $H \subseteq V$ of vertices satisfying the following:

$$H = \underset{U \subseteq V, |U| \leq \kappa |V|}{\arg \max} I((G_U, X_U); C). \tag{1}$$

Here, $G_U$ is the subgraph of $G$ induced by the vertices in $U$, and $X_U$ is the matrix of corresponding feature vectors. The standard definition of function $I(\cdot; \cdot)$ is the *Shannon mutual information* Cover & Thomas (2006), defined as follows: for random variables $X$ and $Y$ on a common probability space,

$$I(X; Y) = \mathbb{E}_{P(X,Y)} \left[ \log \frac{P(X, Y)}{P(X)P(Y)} \right], \tag{2}$$

where the expectation is taken with respect to the joint distribution of $X$ and $Y$. The objective function in (1) i.e. MI between an attributed graph and its class label is defined as follows: An attributed graph consists of both a graph and a collection of node features. Let $g : (0, \infty) \to \mathbb{R}$ be a convex function with $g(1) = 0$. Given graph $G_V$ with fixed set of vertices $V = \{v_1, \ldots, v_k\}$ and features set $X_V = \{X_{v_1}, \ldots, X_{v_k}\}$, the mutual information between $(G_V, X_V)$ and class variable

$C$ with prior probability $\pi_C$ is given by

$$I(G_V, X_V; C) = \mathbb{E}_{P_G, \pi_C}\left[g\left(\frac{P(G_V, X_V, C)}{P(G_V, X_V)\pi_C}\right)\right] = \mathbb{E}_{P_G, \pi_C}\left[g\left(\frac{P(G_V, X_V|C)}{P(G_V, X_V)}\right)\right], \quad (3)$$

where $P_G := P(G_V, X_V)$ and $\pi_C$ is the prior probability of class $C$.

Going back to (1), intuitively, $H$ is a subset of nodes of $G$ with size at most $\kappa|V|$ whose induced attributed subgraph has maximum mutual information with the class label $C$. To justify (1), as we observe in (2) the mutual information measures the nonlinear dependency between input graphs and their class labels. Thus, an optimal subset of vertices that maximizes this measure should preserve classification performance. While this is a very natural formulation of supervised graph summarization, we immediately run into computational complexity difficulties, as our first main result shows.

**Theorem 1** (NP-hardness of supervised graph summarization). *The problem of maximizing $W$ in (1) with knowledge of the data distribution $\mathcal{D}$ is NP-hard.*

*Proof:* We prove this via a reduction from the $k$-clique problem to ours. Given a graph $G$, we would like to determine whether or not it has a $k$-clique. We will do this by constructing an instance of the supervised graph summarization problem above. We let the class label be $C \sim \text{Bernoulli}(1/2)$. We construct a complete graph $G'$ on the same node set as $G$. We assign an event $E_e$ to each (undirected) edge $e$ in $G'$ as follows: if $e$ is in $G$, then we set $E_{v,w}$ to be $E_{v,w} \sim \text{Bernoulli}(p)$, where $p = C/2$, independent of any other edge weight. If, on the other hand, $e$ is not present in $G$, then we set $E_e$ to be $0$. We set all node features to $1$. Note that, with this distribution, we have that the mutual information between the subgraph $H$ induced by any set of $k$ nodes in $G'$ is given by

$$I((H, X_H); C) = I(H; C) = \sum_{v \neq w \in V_H} I(E_{v,w}; C), \quad (4)$$

where $V_H$ is the set of the nodes on subgraph $H$. This is $\binom{k}{2}$ if and only if $H$ is a $k$-clique in $G$, and it is less if there are missing edges. Thus, if we could solve our optimization problem in polynomial time, then we could solve the $k$-clique problem in polynomial time. This completes the proof. $\blacksquare$

In fact, the reduction that we exhibited yields a stronger result: our approximation problem is, in general, NP-hard to approximate within a constant factor of the optimum. This follows from the NP-hardness of approximation of the max clique problem.

Thus, in the search for practical approaches to graph summarization in a supervised setting, we must be more modest in our expectations, and we do not solve the above optimization problem, though it remains the intuitive motivator of our approach. In Section 3, we describe our approach via optimal transport and the framework of OTC.

## 3 SUPERVISED GRAPH SUMMARIZATION VIA OPTIMAL TRANSPORT

In this section, we review the optimal transport summarization framework from Garg & Jaakkola (2019), then propose our supervised method via information-theoretic measures.

### 3.1 PRELIMINARIES: OPTIMAL TRANSPORT ON GRAPHS

We first describe the framework of optimal transport on graphs. Fix a graph $G$ on $n$ vertices $[n] = \{1, 2, ..., n\}$ with edge set $E(G)$. We let $\widehat{E}(G)$ denote the *directed* edge set of $G$, given by

$$\widehat{E}(G) = \{(v, w) \mid \{v, w\} \in E(G)\}. \quad (5)$$

Let $\rho_0, \rho_1$ be two probability distributions on $[n]$, viewed as vectors. Let $c : E(G) \to \mathbb{R}$ be an arbitrary *cost function*. A *flow* on $G$ is a function $J : \widehat{E}(G) \to [0, \infty)$. The *cost* of a flow is defined:

$$W(J) = \sum_{e \in \widehat{E}(G)} c(e)J(e). \quad (6)$$

The *result* of a flow $J$ with initial distribution $\rho_0$ is defined by $R(J) = \rho_0 + F \cdot J$. Finally, the set of flows from $\rho_0$ with the result $\rho_1$ is denoted by $\mathcal{F}_G(\rho_0, \rho_1)$. Next we define the Wasserstein distance between $\rho_0$ and $\rho_1$ which is the minimum cost of any flow transporting $\rho_0$ to $\rho_1$ on $G$ as follows:

$$W_G(\rho_0, \rho_1) = \inf_{J \in \mathcal{F}_G(\rho_0, \rho_1)} W(J). \tag{7}$$

The method proposed in Garg & Jaakkola (2019) takes $G$, $\rho_0$, $c$, and a compression ratio $\kappa$ as input and finds a $\rho_1$ with support size $\leq \kappa |V(G)|$ with minimal Wasserstein distance to $\rho_0$. The summarizing graph $H$ is then the one induced by the vertices in the support of $\rho_1$. The selling point of their method is that prior information about "node/edge importance" can be encoded in $\rho_0$ and $c$. They do *not* formally define any importance measures. They perform experiments in which $\rho_0$ assigns a probability to node $v$ proportional to its degree (so $\rho_0$ is the stationary distribution of the random walk on $G$, if it is ergodic) and $c$ assigns a higher cost to edges connecting nodes with different attributes (all of their experiments are on graphs with categorical attributes). Unlike the framework that we develop in the present paper, their framework is not a priori supervised. Furthermore, no justification is given for their choice of $\rho_0$ and $c$. In addition, their framework ignores dataset-level statistics (since their setting of $\rho_0$ and $c$ is entirely dependent on the individual graph being compressed) and ignores the absence of edges, which can be informative.

### 3.2 Our method: supervised OT via information theoretic measures

In this section, we extend the above framework to the supervised setting. The core idea is to use the training dataset to learn measures of informativeness about the class label of node features and edges. We then use these to construct the initial distribution and cost function, which can then be used in the framework of the previous work applied to the complete graph on $n$ nodes to find a subset $H$ of nodes such that the subgraphs induced by $H$ in the test set are informative about the class label while being a small subset of the original vertex set. We run the OTC algorithm on the complete graph because we wish to take into account the informativeness of the presence/absence of edges. The method proposed in the prior work only does this indirectly. We define the initial distribution $\rho_0$ by:

$$\rho_0(v) = \frac{I(X_v; C)}{\sum_{w \in [n]} I(X_w; C)}, \tag{8}$$

where $I(\cdot; \cdot)$ is the MI between node attributes and class label and defined in (2). Next, we propose the cost function $c(\cdot, \cdot)$ as follows:

$$c(v, w) = D_{KL}(E_{v,w} \mid C = 0 \parallel E_{v,w} \mid C = 1) + R_{v,w}, \quad \text{where} \tag{9}$$

$$R_{v,w} = I(X_v; C \mid X_w) + I(X_w; C \mid X_v), \tag{10}$$

and $D_{KL}(A \| B)$ denotes the $KL$-divergence Cover & Thomas (2006) from a random variable $A$ to a random variable $B$ (See Algorithm 1). The intuition for this definition of $\rho_0$ is that we consider nodes to be important if their features carry a significant amount of information about the class label. The intuition behind the definition of cost $c$ in (9) is as follows: in optimal transport, we want a large amount of flow across an edge if and only if the edge itself is not too informative about $C$, and the feature of one of the vertices incident on the edge is not too informative about $C$, conditioned on the value of the other vertex's feature (in other words, the vertices contain redundant information about $C$). In this case, we want the edge cost to be small.

Running the optimal transport compression routine of OTC on the complete graph $K_n$ with $\rho_0, c, \kappa$ as parameters results in a subset $H$ of nodes of cardinality (approximately) $\kappa n$. To compress a graph $G$ in the test set, we simply take the subgraph of $G$ induced by $H$.

Crucial to the performance of our method is the ability to estimate the mutual information and KL divergence quantities in the expressions for $\rho_0$ and $c$ directly without estimating the distributions Yasaei Sekeh & Hero (2018). We chose EDGE Noshad et al. (2019) because it has been shown that it achieves optimal computational complexity $O(N)$ where $N$ is the sample size. This is significantly faster than its plug-in competitors Kraskov et al. (2004); Moon et al. (2017). It is proved that in addition to fast implementation EDGE has the optimal parametric MSE rate of $O(1/N)$ under a specific condition. The KL divergences are estimated via the plug-in method. In Section 5, we evaluate our method on synthetic and generated graphs from real datasets. We show that it empirically outperforms significantly the previous work on all of the considered datasets, in terms of running time and, on some datasets, in terms of test accuracy.

---

**Algorithm 1:** Supervised Optimal Transport Compression

---

**Data:** Size $n$ of graphs, target subgraph size $k$, dataset $D = \{(G_j, X_j, E_j, C_j)\}_{j=1}^m$, where $G_j$ is an $n \times n$ adjacency matrix, $X_j \in \mathbb{R}^{n \times d}$ is a matrix of vertex features, $E_j$ is a matrix of edge features, and $C_j \in \{0, 1\}$ is a class label

**Result:** A subset $S \subseteq [n]$ of size $k$

```
// Compute initial distribution p_0 on nodes.
```
**1 for** $v = 1$ *to* $n$ **do**
```
    // Compute the MI between the features of node j and the
       class label.
```
**2**     Set $p_0(v) = \frac{I(X_v; C)}{\sum_{w \in V} I(X_w; C)}$;

**3 end**

**4** Set $s = \sum_{j=1}^n p_0(j)$;

**5 for** $j = 1$ *to* $n$ **do**

**6**     Set $p_0(j) = p_0(j)/s$;

**7 end**
```
// Compute n × n cost matrix cost.
```
**8 for** $v \neq w$ **do**

**9**     Set $c(v, w) = D_{KL}(E_{v,w} \mid C = 0 \parallel E_{v,w} \mid C = 1) + (10)$

**10 end**

**11** Set $S = OTCompress(D, p_0, cost)$;

**12 return** $S$;

---

## 4    LIMITATIONS OF THE OPTIMAL TRANSPORT APPROACH

In this section, we explore the limitations of the optimal transport approach for the purposes of maximizing the mutual information of graphs $(G, X)$ with a class label $C$. To solve this optimization problem via OT we require initial distribution $\rho_0$ and cost function $c$. Here, we first specify the kinds of initial distribution and cost function we propose in (8) and (9).

**Definition 1** (Data distribution-dependent parameters). *Fix a graph size $n$. An edge cost function is a function $c$ from the joint distribution of $(G, X_V, C)$ (call this the **data distribution**) and pairs of vertices (i.e., numbers in $[n]$) to non-negative real numbers or $\infty$. If the data distribution is given, then $c$ is just a mapping from pairs of vertices to $[0, \infty]$.*

*Similarly, an initial distribution map $L$ is a mapping from the data distribution to probability distributions on $[n]$ (which are meant to be values for $p_0$). If the data distribution is given, then $p_0$ is a function mapping vertices to probabilities (which must sum to 1).*

Below we define a property of OT parameter pairs called *information monotonicity*, which guarantees that using a pair with this property, flow cost decreases as the mutual information between the resulting compressed subgraph and the class label increases. We will study to what exent OT parameter pairs satisfy this property.

**Definition 2** (Information monotonicity). *An OT parameter pair $(c, L)$ is said to be* information monotone *for a data distribution $(G, X_V, C)$ if, for any two flows $f^{(0)}, f^{(1)}$ on $G$ resulting in subgraphs $H^{(0)}, H^{(1)}$, respectively, we have*

$$W(f^{(0)}) \leq W(f^{(1)}) \iff I((H^{(0)}, X_{H^{(0)}}); C) \geq I((H^{(1)}, X_{H^{(1)}}); C). \tag{11}$$

### 4.1    A MONOTONICITY COUNTEREXAMPLE

In this section, we construct an example graph for which the Wasserstein distance between $p_0$ and $p_1$ is not monotone with respect to the mutual information between the graph and the class label random variable. Let us suppose that all vertex features are independent of the graph structure and of the class label, so they provide no information about either. In this case, $p_0$ is the uniform distribution over all $n$ vertices. Furthermore, let's suppose that the edges are generated as follows: the class label $C \sim \text{Bernoulli}(p)$, and then, conditioned on $C = c$, each edge $e$ appears with probability

$q_e^{(c)}$, independent of everything else. Intuitively, those edges with large $|q_e^{(0)} - q_e^{(1)}|$ provide more information about the value of $C$.

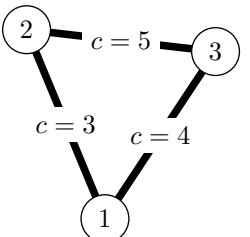

Set $n = 3$, so we have a triangle. We can cook up settings of $q_e^{(c)}$ such that the costs of the three edges are proportional to $3, 4, 5$ i.e we have the cost function $c(1,2) = 3, \ c(1,3) = 4, \ c(2,3) = 5$ scaled appropriately.

To be more explicit about the calculation of the cost of one edge $e$, we write:

$$c(e) = D_{KL}(E_e \mid C = 0 \parallel E_e \mid C = 1) = q_e^{(0)} \log \left( \frac{q_e^{(0)}}{q_e^{(1)}} \right) + (1 - q_e^{(0)}) \log \left( \frac{1 - q_e^{(0)}}{1 - q_e^{(1)}} \right). \quad (12)$$

(Note that the $R_e$ term vanishes in our example because of our independence assumption.) The cost $c(e)$ can take any value between $0$ and $\infty$, if suitable values of $q_e^{(0)}$ and $q_e^{(1)}$ are chosen. We thus can cook up choices of these parameters that give exactly the values $3, 4, 5$. Let us consider two flows:

- In one, we set $J(1,2) = J(1,3) = 1/6$ and set the flow across all other edges to $0$. The cost of this flow is $3 \cdot 1/6 + 4 \cdot 1/6 = 7/6$. The distribution $p_1 = (0, 1/2, 1/2)$. This leaves in the compressed graph the edge $\{2, 3\}$, which had cost $5$ (so, the highest possible mutual information with the class label).
- In the other, we set $J(2,1) = 1/3$ and set all other flows to $0$. The cost of this flow is $3 \cdot 1/3 = 1$. The distribution $p_1 = (2/3, 0, 1/3)$. This leaves the edge $\{1, 3\}$, which had cost $4$ (so, *not* the highest possible mutual information with the class label).

**Discussion:** In this counterexample, we used a specific choice of cost function and showed that there is a fundamental limitation of the OT approach in Garg & Jaakkola (2019) for classification problems.

Now, we have the following theorem regarding non-monotonicity of optimal transport solutions with respect to mutual information, which generalizes the above example.

**Theorem 2** (Non-monotonicity ). *Let $c(\cdot, \cdot)$ be a cost function satisfying the following properties:*

• *$c(u, v)$ is strictly monotone increasing with the mutual information between $E_{u,v}$ and $C$ and is $0$ when the two are independent.*
*Let $L$ be an initial distribution map satisfying the following properties:*

• *if $X_V$ is independent of $C$, both unconditionally and conditioned on $G$, then $p_0$ is the uniform distribution.*

*Then there exist data distributions for which $(c, L)$ is not information-monotone.*

*Proof:* We will prove this by explicit construction. We first describe the data distribution that leads to our counterexample: Let us suppose that all vertex features are independent of the graph structure and of the class label, so they provide no information about either. In this case, $p_0$ is the uniform distribution over all $n$ vertices. Furthermore, let's suppose that the edges are generated as follows: the class label $C \sim \mathrm{Bernoulli}(p)$, and then, conditioned on $C = c$, each edge $e$ appears with probability $q_e^{(c)}$, independent of everything else. Intuitively, those edges with large $|q_e^{(0)} - q_e^{(1)}|$ provide more information about the value of $C$ (this is true regardless of what the cost function is). Let $e_*$ denote the edge for which $|q_{e_*}^{(0)} - q_{e_*}^{(1)}|$ is maximized. Suppose that the vertices of $e_*$ are $v_0, w_0$. Let $e = \{v_1, w_1\}$ denote some arbitrary non-information-optimal edge. We will construct two flows: $f^{(0)}$ will move all probability mass to $e_*$, while $f^{(1)}$ will move all mass to $e$.
**(i)** for any $u \notin \{v_0, w_0\}$, $f_0$ sends all $1/n$ of the mass on $u$ to $z_0(u) = \underset{x \in \{v_0, w_0\}}{\arg\min} \ c(u, x)$.

**(ii)** for any $u \notin \{v_1, w_1\}$, $f_1$ sends all $1/n$ of the mass of $u$ to $z_1(u) = \underset{x \in \{v_1, w_1\}}{\arg\min} \ c(u, x)$.

The resulting total cost of $f^{(b)}$, for $b \in \{0, 1\}$, is $W(f^{(b)}) = \frac{1}{n} \sum_{u \notin \{v_b, w_b\}} c(u, z_b(u))$.

Now, it remains to show that $W(f^{(0)}) \geq W(f^{(1)})$, for *some* choice of model parameters $q_e^{(b)}$,

$b \in \{0, 1\}$. This is easy to guarantee since the edge sets of which the two sums indexed are disjoint, and we can make the costs arbitrarily close to $0$ for the edges in $f_1$ by decreasing $|q_{u,x}^{(0)} - q_{u,x}^{(1)}|$ sufficiently. Since we only decrease this quantity for edges that are not $e_*$, this does not affect the optimal edge. This completes the proof.

The above counterexample suggests the following intuition behind the shortcomings of the optimal transport approach: an edge may be very informative, but it may be isolated from the rest of the graph by high-cost (informative) edges. That is, the informativeness of an edge may have nothing to do with the cost of its surrounding edges. In this case, optimal transport might find it hard to pick it out, simply because there is a lower cost associated with flowing elsewhere. Hence, there's a tension: if an informative edge has a high cost, then it may be costly to flow to it. If an informative edge has a low cost, then it may be too cheap to direct flow out of it.

## 5 EXPERIMENTAL STUDY

We conducted several experiments to compare the performance of our method with that of the OT approach and baseline (no-compression). We use graph summarization methods including OTC, Ours (Algorithm 1), and Coarsening(GC) Jin et al. (2020) on each graph classification dataset, then use the resulting summarized dataset to train and test a classifier as described later in this section. We evaluate our work using both real and synthetic datasets as described below. All the datasets have binary class labels. Note that we first identify class-sensitive nodes for all datasets by using the measure $\rho := \mathbb{E}_{(X_V, C) \sim D}[P(X_V|C)/P(X_V)]$ where $P(X_V|C)$ and $P(X_V)$ are conditional and marginal density functions, respectively. $\rho$ determines how the distribution of node features changes given class labels. This way we focus on nodes whose features carry more information for class labels to deliver further emphasis on the contribution of both features and edges in graph classification. The node size in Table 1 is after applying $\rho$.

**Image-based Dataset:** The real vision datasets that we are using here are based on MNIST Deng (2012), CIFAR-10 Krizhevsky (2009) and MiniImageNet Vinyals et al. (2016) datasets. Each pixel in an image is assigned to a node in a graph and each image is a graph of attributed nodes where the edges between the nodes are based on their spatial relationships: adjacent pixels are linked by an edge similar to Defferrard et al. (2016). It is notable that the number of nodes for each graph would be tremendous since images are $784 \times 784$ pixels and we have a huge adjacency matrix for each graph. So, compressing the nodes here would be necessary specifically when considering a lot of non-useful pixels that might even negatively impact the classification.

**Transport-based Dataset:** As a non-vision real dataset, the New York Taxi Data (2010-2013) Donovan & Work (2016) is used to generate an attributed graph dataset with a fixed node size among all samples. In the raw data, there are about 7 million trips by Taxis in New York City (NYC), and 703 graph samples are extracted out of them with the pre-processing as the following. Firstly, invalid trips are eliminated and then the "vendor_id"s are considered as the labels. So, the trips are divided into two categories based on their vendor ids. In each category, the trips of each day are considered as one single sample, and the nodes, edges, and nodes' attributes are generated for each sample. For generating the nodes, each element of a grid over the NYC map is assigned to a node. So, each node in each sample shows a neighborhood on the map in a day. The attributes of these nodes are calculated as the summation of the time that taxis spend on a trip when they pick up and drop off people only in that neighborhood. There is an edge between two nodes if there was at least a trip on that day between those two nodes.

**Synthetic Dataset:** The invoked synthetic data has class-label-dependent node features and edges. The feature of node $j$ is generated randomly from a Gaussian distribution with standard deviations and means $j/|V| \times 5$ for class $0$ and $j/|V| \times 4$ for class $1$. Edges are randomly added to the graphs using the probability assigned to the class label.

**Experimental design:** For the classification accuracy experiments, for each dataset and method, we train the summarization method on a training set, then apply the summarization method with the learned parameters to a test set. We then use the resulting summarized dataset as input to a classification pipeline: this pipeline is trained/tested using 5-fold cross-validation. We report the average and standard deviation of the test accuracy over 5 trials for each compression and classifier. We also report the running time of the training (when applicable) and testing phase of the compression

time, as well as the running time of the classification pipeline on the summarized dataset. We note that the latter running time is important because one of the central purposes of graph summarization is to make subsequent tasks, such as classification, more computationally efficient while not sacrificing too much accuracy. These statistics are *not* reported in prior work Garg & Jaakkola (2019). The same experiment is performed with multiple compression ratios. For the classification model, we apply a support vector machine (SVM) with the GraphHopper kernel Feragen et al. (2013). We use this kernel because of its ability to handle continuous node attributes.

## 5.1 PERFORMANCE ANALYSIS ON GRAPH CLASSIFICATION DATASETS

To evaluate our method and analyze its performance we compared the test accuracy of our method, OTC, Graph Coarsening (GC) Jin et al. (2020), and baseline (i.e. no compression). Among other methods, we focused on only Graph Coarsening and OTC which already outperforms several existing works. Table 1 summarizes the performances for each compression ratio. As the numbers in bold indicate, our method outperformed the other methods across the datasets on most compression ratios in expectation. Our method outperforms OTC, GC, and baseline on MNIST, NYC and Synthetic datasets for all compression ratios while for CIFAR10 and miniImageNet we could beat OTC on some of the ratios (0.3-0.6 for CIFAR10 and 0.6, 0.7 for miniImageNet).

Table 1: This table demonstrates the accuracy of OTC, Ours, and Graph Coarsening(GC) Jin et al. (2020) methods on several real and synthetic datasets for some compression ratios. The methods with the best performance are indicated in bold font in each case. Here Ours+($\rho$) is the experiment to improve "Ours" accuracy. For fairness "Ours" is the result that should be compared against OTC.

| Dataset | Method | Compression Ratios | | | | | | | |
|---|---|---|---|---|---|---|---|---|---|
| | | 0.2 | 0.3 | 0.4 | 0.5 | 0.6 | 0.7 | 0.8 | 1.0 |
| **MNIST** | GC | .538±.066 | .496±.068 | .492±.081 | .558±.082 | .571±.070 | .542±.070 | .571±.081 | - |
| Graphs:300 | OTC | .687±.113 | .633±.123 | .7±.151 | .673±.139 | .787±.167 | .76±.122 | .747±.128 | - |
| Nodes: 100 | Ours | **.937±.061** | **.945±.078** | **.939±.095** | **.927±.115** | **.917±.118** | **.883±.136** | **.836±.169** | - |
| Edges: 179.8 | Baseline | - | - | - | - | - | - | - | .793±.017 |
| **CIFAR10** | GC | .492±.002 | .492±.002 | .506±.006 | .506±.006 | .493±.005 | .493±.005 | .506±.006 | - |
| Graphs: 600 | OTC | **.516±.021** | .51±.027 | .493±.029 | .493±.039 | .503±.046 | .517±.035 | **.53±.027** | - |
| Nodes: 200 | Ours | .501±.026 | **.541±.031** | **.549±.038** | **.549±.022** | **.532±.033** | .53±.026 | .521±.021 | - |
| Edges: 335.15 | Ours+($\rho$) | .507±.031 | .522±.033 | .536±.032 | .53±.034 | .528±.030 | **.536±.030** | .528±.026 | - |
| | Baseline | - | - | - | - | - | - | - | .53±.012 |
| **MiniImageNet** | GC | .493±.003 | .504±.006 | .493±.003 | .493±.003 | .557±.003 | .504±.006 | .504±.006 | - |
| Graphs: 1000 | OTC | .573±.027 | **.576±.022** | **.583±.020** | .564±.037 | .559±.041 | .569±.040 | **.571±.029** | - |
| Nodes: 250 | Ours | .565±.039 | .562±.043 | .568±.042 | .561±.041 | **.569±.038** | **.569±.039** | .567±.043 | - |
| Edges: 111.06 | Ours+($\rho$) | **.58±.029** | .57±.038 | .566±.043 | **.566±.042** | .569±.039 | .568±.038 | .57±.036 | - |
| | Baseline | - | - | - | - | - | - | - | .569±.028 |
| **NYC** | GC | .524±.048 | .529±.077 | .522±.037 | .520±.064 | .510±.088 | .525±.062 | .5256±.041 | - |
| Graphs: 703 | OTC | N/A | .874±.051 | .922±.031 | **.952±.02** | .918±.025 | .888±.03 | **.918±.038** | - |
| Nodes: 100 | Ours | N/A | **.96±.017** | **.955±.020** | .943±.026 | **.929±.031** | **.907±.047** | .889±.059 | - |
| Edges:: 418.13 | Baseline | - | - | - | - | - | - | - | .871±.04 |
| **Synthetic** | GC | .625±.026 | .655±.053 | .656±.013 | .708±.031 | .732±.035 | .734±.026 | .763±.027 | - |
| Graphs: 1500 | OTC | .886±.039 | .918±.016 | .957±.011 | .966±.013 | .984±.005 | .986±.004 | .992±.003 | - |
| Nodes: 100 | Ours | **.909±.014** | **.950±.01** | **.981±.005** | **.984±.007** | **.986±.014** | **.994±.003** | **.993±.004** | - |
| Edges:4017.41 | Baseline | - | - | - | - | - | - | - | .992±.005 |

It is notable that we outperform other methods' test accuracy in half of the real datasets, along with the synthetic dataset, by a large margin. We hypothesize that this is not happening for CIFAR10 and MiniImageNet because our chosen optimal transpose parameters only incorporate the local informativeness of each node. In such complex image classification tasks, pixels in isolation are less informative than in global image structure. This may result in our method failing to perform

significantly better than the baseline in terms of test accuracy. To improve the accuracy of our method for CIFAR10 and miniImageNet, we have run another experiment using again $\rho$ (called "Ours+$\rho$ on Table 1). We succeed on $0.7$ ratio for CIFAR10and$0.2$, $0.5$ ratio for miniImageNet. *Note that for the sake of fairness "Ours" is the result that should be compared against OTC* as the sensitive nodes are selected only once in OTC.

As observed in the table for most of the examined datasets, the standard deviations are sufficiently large similar to the other well-known graph compression methods– that we cannot make empirical high-probability statements of superiority over other methods. Here "baseline" means a classification performance on a non-summarized graph.

Furthermore, classification error is not the only important evaluation metric; training and testing speed is also essential. Our method is uniformly superior to the baseline in terms of these metrics, because the baseline performs the optimal transport summarization on each new graph, whereas in our method this is only done on the training set. Figure 1 demonstrates that our method outperforms OTC with a very large margin in terms of compression time. It shows that our method outperforms both OTC and baseline (no compression) in terms of classification times. This is because they find the best-compressed vertex set for each sample individually while we find a single best-compressed vertex set for the whole sample.

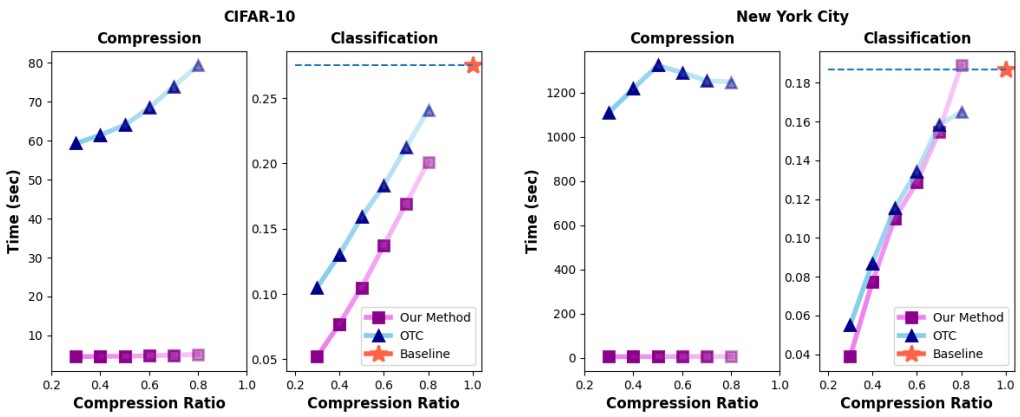

Figure 1: The compression and classification time for both Ours and OTC methods is shown for both CIFAR-10 and NYC. Our method substantially outperforms OTC and baseline (no compression) in both classification and compression times. The classification time for the compression ratio with the best accuracy for our method is less than half of that of the baseline.

## 6 Conclusion

We have introduced a theoretical formulation of supervised graph summarization in terms of maximization of the mutual information between the class labels and graphs to be classified. We showed that the solution to this problem, even approximately, is NP-hard, and so we took a different approach. In particular, we showed how the unsupervised optimal transport graph summarization framework of Garg & Jaakkola (2019) can be adapted – nontrivially – to a supervised setting via estimation of information-theoretic measures incorporating both graph structure and node features. This is in contrast to the parameter settings of the previous work, which only took into account the degree distribution of the graph and compressed graphs in isolation, failing to take into account dataset-wide statistics. We also elucidated limitations of the optimal transport solution for maximization of the mutual information objective in our summarization framework.

**Looking ahead** - Further work is necessary to elucidate the precise power/limitations of the optimal transport approach to graph summarization. For instance, its connections to other methods, such as graph clustering-based methods deserve further examination. Moreover, our adaptation of the optimal transpose method to the supervised setting requires further development to accommodate graph datasets in which different graphs do not share a common vertex set.

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
