# OpenReview forum: "Optimal Transport-Based Supervised Graph Summarization"
_ICLR.cc/2023/Conference — Submitted to ICLR 2023_

### Official Review · Reviewer_MXgV · 2022-10-20

**Confidence:** 2
**Correctness:** 4
**Technical Novelty And Significance:** 2
**Empirical Novelty And Significance:** 2
**Recommendation:** 5

**Clarity, Quality, Novelty And Reproducibility:**

The clarity of the paper could be improved by a large margin by giving more insights on intuitions behind the exposed claims / theorems and reorganizing the paper structure such that the whole message is more clearly conveyed.

**Strength And Weaknesses:**

* Strengths
    * The use of supervised information to guide the summarization process makes sense
    * Relying on Information-theoretic measures for that seems like a reasonable choice
    * The study of the inherent limitations of OT-based approaches for graph summarization is interesting
* Weaknesses
    * The contribution in terms of method is rather limited
    * The paper is presented in such a way that it makes it hard to grasp the repercussions of the findings

In more details:

* Regarding the limited contribution
    * The contribution focuses on setting node weights $\rho_0$ and cost based on an information-theoretic criterion, rather than based on vertex / edge importance, as done in the OT baseline, which is reasonable yet not ground-breaking.
    * In terms of performance, the average gain seems rather limited when compared to the reported standard deviations, except on the Synthetic dataset, which is build on purpose to match the method's specificities. This is rather disappointing given that the OT baseline uses no information for the graph summarization step.
* Regarding the lack of clarity
    * In general, the paper lacks both structure and discussions about the potential impacts of the findings. The paper should be largely rewritten to improve on understandability.
        * Regarding Structure, as an example, Section 5 is interesting but it misses an important point which is to discuss whether the proposed method (Section 4.2) suffers from the limitations presented in Theorem 2 or not
            * This is discussed in the experimental section (6.2) but it would be nice to provide insights on this question in Section 5
        * Regarding discussions, here are a few examples:
            * In Section 4.3, the assumption "that the random node attributes are independent" seems rather strong and not sufficiently motivated. Also, I do not understand the formulation in Equation (13): could the authors detail their derivation for $P(G_V | X_V, C)$ and the underlying assumptions if there are any?
            * In Section 5, the counter-example should be better discussed so as to convey the important message about why this specific example is a counter example, rather than focusing on calculations. Maybe starting with the Theorem + proof and then using the example could avoid redundancy in the text and help focus the counter-example on its illustrative power


* Minor issues
    * Strange that authors do not cite anything related to Gromov-Wasserstein for graphs in their literature about the use of Optimal Transport on graphs
    * Section 2 is a copy-paste of the beginning of Section 3!
    * Theorem 2: $X_G$ => $X_V$?
    * I do not understand what the authors mean by "we tag best out of XX% of nodes as sensitive using ..."

**Summary Of The Paper:**

This paper introduces a variant over the OT-based graph summarization technique introduced in (Garg & Jaakola, 2019). The improvement relies on taking supervised information into account (when it exists) through the use of Mutual Information.

**Summary Of The Review:**

The paper proposes a method to use supervised information for graph summarization using Optimal Transport.
The proposed method is of limited scope (even experiments do not clearly illustrate its interest) and the presentation is not sufficiently clear to convey the overall message.

---

> ### Author Response · Authors · 2022-11-19
> **Response to Reviewer MXgV**
>
> We really appreciate your review on our work and we thank you for your time and careful evaluation of our paper. Please see below our responses to your comments/questions:
>
> - The contribution in terms of method is rather limited,
> - The paper is presented in such a way that it makes it hard to grasp the repercussions of the findings,
>
> **Response:** We acknowledge the comment and we have fixed this in the revision.
>
> - Regarding the limited contribution
>
> **Response:**
> 1. We want to emphasize that while the baseline paper mentions vertex/edge importance, they never define these concepts, and they don't give or use any objective measures of ``importance''.  For instance, the node weights $\rho_0$ in the previous work are set to be simple functions of the degrees of the nodes.  These choices were not accompanied by any intuitive or formal justification.
> 2. We note that there are a few real datasets listed in our experiments, on which our method outperforms the baseline. For a further discussion of possible explanations for the observed behavior in our experiments, please see our response to Reviewer atLk.
>
>     We also note that, conditioned on the class label, the graph model that was used to derive the graph mutual information estimator has precedent --  it is a generalized version of a latent position vector model in which the latent positions are independent.  This is an assumption that has appeared in empirical literature before (e.g.,  https://arxiv.org/pdf/1709.05506.pdf).  However, it is clearly not always an appropriate assumption and likely does not hold for image data, for instance.
>
> - Regarding the lack of clarity
>
> **Response:**
> 1. We believe we have fixed this issue in the revision.
>     - Regarding Structure: The proposed method does, indeed, suffer from the limitations presented in Theorem 2.  If one wanted to surmount the limitations presented in Theorem 2, it seems that one must take into account global graph structure -- in particular, clusters of informative nodes and edges.  We are not certain that this can be done in such a way that the resulting method simultaneously yields substantial performance advantages and maintains the spirit of the optimal transport approach (which is to say, deals with optimizing network flows).
>     - Regarding discussions:
>
>         1. Regarding the assumption about node attributes being independent, please see our discussion of latent position vector models above.  An assumption, which is a component of such models, but which we forgot to mention, is that conditioned on the incident vertex attributes, the presence or absence of an edge is independent of everything else.  However, we stress that this assumption is only needed for the additional experiments in supplementary materials (SM) and graph mutual information estimator (provided in the SM) and not for our compression method.
>         2.  Regarding the counter-example, since we are using its information in the theorem and proof, we must define it before them. Also, we have mentioned that "an edge may be very informative, but it may be isolated from the rest of the graph by high-cost (informative) edges. That is, the informativeness of an edge may have nothing to do with the cost of its surrounding edges." that says why it is a counter-example.
>  2. Minor issues: We have fixed these issues in the revision.
>  3. Clarity, Quality, Novelty And Reproducibility: We have fixed these issues in the revision.
>
> - Please let us know if you have any questions.

---

> > ### Comment · Reviewer_MXgV · 2022-11-24
> > **Response to authors**
> >
> > I would like to thank the authors for this valuable feedback.
> >
> > Though I agree that the form of the paper is a bit better, I believe the paper still needs more work to be more accessible to the reader.
> > I hence decide to change my rating from Reject to Weak reject.

---

### Official Review · Reviewer_9TND · 2022-10-24

**Confidence:** 3
**Correctness:** 2
**Technical Novelty And Significance:** 2
**Empirical Novelty And Significance:** 1
**Recommendation:** 3

**Clarity, Quality, Novelty And Reproducibility:**

- The paper's writing and presentation is unclear and confusing. I could not understand its description of previous work, and more seriously, it's problem setting.
- Section 2 and the early part of Section 3 are repetitive, even with that, I could not understand its content. There are things like "D" without any connections to the already confusing text.
- Objective function (1) 3) are unclear.

**Strength And Weaknesses:**

+The problem motivation is valid (if the solution is available)
- The paper's writing is too unclear and erroneous to the level that I could not follow.

**Summary Of The Paper:**

The paper seems to work on graph summarization problem taking into account node attribute and class label, hence, supervised graph summarization.

**Summary Of The Review:**

The paper didn't have a proper presentation that allows readers to follow, let alone judging its quality.

---

> ### Author Response · Authors · 2022-11-19
> **Response to Reviewer 9TND**
>
> We really appreciate your review on our work and we thank you for your time and careful evaluation of our paper. Please see below our responses to your comments/questions:
>
> - The paper's writing is too unclear and erroneous to the level that I could not follow.
>
> **Response:**  Thank you for the feedback. We have proofread and clarified the problem motivation and contribution.
>
> - The paper's writing and presentation is unclear and confusing. I could not understand its description of previous work, and more seriously, it's problem setting.
>
> **Response:**  We acknowledge that our initial description of our contributions needs substantial revision for clarity and this is addressed in the revision. However, we would appreciate it if you could please be more specific on what was unclear in the description of previous work and in the problem statement.  As far as we can tell, the problem statement is rigorous, including the objective function in equation (1).  It is the subset $W$ of nodes that maximizes the mutual information between the ordered pair $(G_W, X_W)$ and the class label $C$ of the graph (both this ordered pair and $C$ are random variables on the same probability space, so it makes sense to talk about their mutual information).  Here, $G_W$ is the induced subgraph on the nodes $W$, and $X_W$ is the accompanying collection of node attributes.  The subset $W$ is constrained to have cardinality less than or equal to $\kappa \cdot |V|$.  Is it perhaps the notion of the mutual information between an ordered pair and a different random variable that confuses you?  This is a mathematically well-defined notion, but we can clarify more if that is helpful.
>
> We do see several parts where the writing can be improved, and we did so in the revision.
>
> -  Section 2 and the early part of Section 3 are repetitive, even with that, I could not understand its content. There are things like "D" without any connections to the already confusing text.
>
> **Response:**  We have fixed this issue in the revision.
>              We also agree that ``D'' was confusing and we have fixed this by using ${(G_i,X_i,C_i)}_{i=1}^m$ to avoid confusion.  Note that $\mathcal{D}$ is defined to be the data distribution.
>
> - Objective function (1) 3) are unclear.
>
> **Response:** Please see our revision for clarification.
>
> - Please let us know if you have any questions.

---

> > ### Comment · Reviewer_9TND · 2022-11-24
> > **Still hard to read.**
> >
> > - W did not appear in (1), maybe you meant U in (1)?
> > - "Thus, an optimal subset of vertices that maximizes this measure should preserve classification performance."- How can you prove this (if it makes sense at all)?
> > - "While this is a very natural formulation of supervised graph summarization ... "
> > - Theorem 1: "with knowledge of the data distribution D". This contradict the setting of (1) "no knowledge of distribution D". Not sure what is and where knowledge of D is used in the theorem.
> > - "The result of a flow J with initial distribution ρ0 is defined by R(J) = ρ0 + F · J." F is undefined, so is the dot product with function J.
> > - H is confusing: "... select a subset H ⊆ V of vertices satisfying ... ", "... the subgraph H induced by any set of k nodes in..." ....
> > - V_H seems to mean both the case of H as a set of vertices as well as a subgraph. This should be stated.
> > - "We run the OTC algorithm on the complete graph because we wish to take into account the informativeness of the presence/absence of edges." The role of the complete graph K_n is not shown anywhere. I thought OTC is run on the set of training graphs to obtain a subset of vertices H?
> >
> > With help of the response and more reading of the paper plus guessing, I could barely get the idea of the paper, that is to have prior distribution and cost of transportation that are relevant to the label? The idea can be useful, but it is necessary to have a clean description of the idea and why it makes sense to do so. I appreciate the response, but I could not see a paper in this current version.

---

> > > ### Author Response · Authors · 2022-12-08
> > > **Responses to Questions/Comments**
> > >
> > > Thank you very much for taking the time to provide us with your questions and concerns regarding the paper. We've addressed them and we hope that our responses found below help clarify our work.
> > >
> > > **1. W did not appear in (1), maybe you meant U in (1)?**
> > >
> > > In (1) in the revised version of the paper for the sake of consistency we have used H for the subset of nodes with maximum mutual information.  However, in our response the goal was to explain the problem statement therefore due to *argmax* function in (1), using W or H would provide the same clarification.
> > >
> > > **2. "Thus, an optimal subset of vertices that maximizes this measure should preserve classification performance."- How can you prove this (if it makes sense at all)? "While this is a very natural formulation of supervised graph summarization ... "**
> > >
> > > We would like to point out that while this is not true in general, the use of mutual information maximization has precedent in supervised feature selection, so much so that it has been given the name *InfoMax* principle. In learning problems, InfoMax is a well-known approach that has been studied extensively. For instance, in [1-3], the authors propose Deep Infomax (DIM) and investigate unsupervised/semi-supervised learning of representations by maximizing mutual information between an input and the output of a deep neural network encoder and in [4], the authors present Deep Graph Infomax (DGI), a general approach for learning node representations within graph-structured data in an unsupervised manner.
> > >
> > > [1]. Devon Hjelm, et al. LEARNING DEEP REPRESENTATIONS BY MUTUAL INFORMATION ESTIMATION AND MAXIMIZATION, ICLR 2019.
> > >
> > > [2]. Sun et al. INFOGRAPH: UNSUPERVISED AND SEMI-SUPERVISED GRAPH-LEVEL REPRESENTATION LEARNING VIA MUTUAL INFORMATION MAXIMIZATION, ICLR 2020.
> > >
> > > [3]. Tschannen et al. ON MUTUAL INFORMATION MAXIMIZATION FOR REPRESENTATION LEARNING, ICLR 2020
> > >
> > > [4]. Velickovic  et al. DEEP GRAPH INFOMAX, ICLR 2019
> > >
> > > **3. Theorem 1: "with knowledge of the data distribution D". This contradict the setting of (1) "no knowledge of distribution D". Not sure what is and where knowledge of D is used in the theorem.**
> > >
> > > This is not a contradiction.  We are saying that the problem is hard even when we give ourselves knowledge of the data distribution $D$.  So in a setting where we have no knowledge of the data distribution, of course, the problem is even harder.
> > >
> > > **4."The result of a flow J with initial distribution ρ0 is defined by R(J) = ρ0 + F · J." F is undefined, so is the dot product with function J.**
> > >
> > > Here we use the same notations as in Garg & Jaakkola (2019): The function $J$ can be viewed as a vector with each entry corresponding to an ordered pair of distinct vertices in the graph. The matrix $F$ is the signed incidence matrix of the graph, whose rows are indexed by ordered pairs of vertices, and whose columns are indexed by vertices: i.e., $F(\vec{e}, v) = 1$ if $\vec{e} = (w, v)$ is an edge in the graph, for some vertex $w$, and $F(\vec{e}, v) = -1$ if $\vec{e} = (v, w)$ is an edge in the graph for some $w$, and $0$ otherwise.  Furthermore, $F$ should be transposed in the given formula for $R(J)$.
> > >
> > > **5. H is confusing: "... select a subset H ⊆ V of vertices satisfying ... ", "... the subgraph H induced by any set of k nodes in..." .... V_H seems to mean both the case of H as a set of vertices as well as a subgraph. This should be stated.**
> > >
> > > Thank you for pointing this out. As seen in the graph theory literature, subgraphs and the nodes comprising them are frequently given the same notation and this abuse of notation is quite common in graph literature.
> > >
> > > **6. We run the OTC algorithm on the complete graph because we wish to take into account the informativeness of the presence/absence of edges." The role of the complete graph K_n is not shown anywhere. I thought OTC is run on the set of training graphs to obtain a subset of vertices H?**
> > >
> > > We agree that understanding this is a necessary condition for understanding our method. Let us explain it: In the prior work, running OTC on each individual graph does not allow for the incorporation of dataset-wide statistics regarding the informativeness of edges for classification.
> > >  Our method is meant to correct for this by assigning an initial distribution and edge weights that are functions of the amount of information that the presence/absence of each edge gives about the class label.  Since *every* potential edge (i.e., distinct pair of vertices) may be informative about the class label, we must consider all potential edges, and this is why we run OTC on a complete graph.
> > >
> > >
> > > We appreciate the opportunity to help clarify any questions/comments and we would be grateful if you'd take a look at our responses to other reviewers. We hope our clarifications have cleared your concerns, but if you still have any questions we would gladly address them.

---

> > > > ### Comment · Reviewer_9TND · 2022-12-09
> > > > **Thanks for the explanations**
> > > >
> > > > I've shown some of the difficulty I faced when reading the paper. The explanations partly help, but I'd prefer the paper clear and precise on its own.

---

### Official Review · Reviewer_u1qZ · 2022-10-25

**Confidence:** 3
**Correctness:** 3
**Technical Novelty And Significance:** 3
**Empirical Novelty And Significance:** 2
**Recommendation:** 6

**Clarity, Quality, Novelty And Reproducibility:**

Overall this is a neat, original work well written, however it's limited in the scope as mostly an extension of a previous work.

**Strength And Weaknesses:**

Overall this is a neat extension of Garg & Jaakkola (2019) which proposed to address graph summarization from optimal transport point of view. The formation of mutual information under the existing optimal transportation is novel.

While the work overall is a good one, this works' contribution, however,  is limited in that it mostly extends Garg & Jaakkola (2019). Although the theoretically information NON-monotonicity is pointed out, it's not sure how this empirically affects the summarization. Furthermore the experiments would be better improvised with further datasets (e.g. several standard graph datasets in  Garg & Jaakkola (2019)) and visualizations.


**Summary Of The Paper:**

In this work the authors address the problem of graph summarization using optimal transport with class labels in mind. In doing so the author proposes to formulate the problem as an optimal transport that mutual information about a class label is maximized. Empirically the author conducts several experiments to show the improvement over the previous work.

**Summary Of The Review:**

Based on the strengths and weaknesses, I would recommend 6: marginally above the acceptance threshold

---

> ### Author Response · Authors · 2022-11-19
> **Response to Reviewer u1qZ**
>
> We really appreciate your review on our work and we thank you for your time and careful evaluation of our paper. Please see below our responses to your comments/questions:
>
> - While the work overall is a good one, this works' contribution, however, is limited in that it mostly extends Garg \& Jaakkola (2019).
>
> **Response:** We note that while our proposed method does build on Garg \& Jaakola (2019), we view our additional contributions as a probing of the fundamental limitations of the optimal transport-based approach for supervised graph summarization, as given in our theoretical and some of our experimental results.  The work of Garg \& Jaakola (2019) is published at a top conference, and so we view critiques of its approach as scientifically valuable.
>
> - Although the theoretically information NON-monotonicity is pointed out, it's not sure how this empirically affects the summarization.
>
> **Response:** Indeed, this is a good question, and one that we hoped to tackle in the future.
>         The theoretical ramification of information non-monotonicity is that there exist certain data distributions for which the mutual information measure does not monotonically increase as the flow cost decreases.  Empirically, it is important to investigate how common these distributions are (in other words, do they arise in practice?), how badly non-monotone they are, and how much this affects classification test accuracy.  Ideally, we would have included these experiments in this paper.
>
> - Furthermore the experiments would be better improvised with further datasets (e.g. several standard graph datasets in Garg & Jaakkola (2019)) and visualizations.
>
> **Response:** You are correct that it would be useful to have additional experiments.  However, one limitation of our current approach is that we must assume that every input graph is defined on the same vertex set (otherwise, it doesn't make sense to talk about the mutual information between $X_v$ and the class label).  Many of the datasets that Garg \& Jaakola (2019) considered did not satisfy this constraint, which is why we did not include them.  This limitation of our work was explained at least in the Conclusion section (also stated, but perhaps not emphasized enough, in the problem formulation section).  We emphasize that there do exist datasets that satisfy our constraint.  We also have some notion of how to extend our method beyond this constraint: instead of considering the mutual information between $X_v$ and the class label, one first maps  nodes of distinct graphs to a common embedding space.  Further work is needed to complete this idea.
>
> - Please let us know if you have any questions.

---

> > ### Comment · Reviewer_u1qZ · 2022-11-24
> > **Repose to Author Response**
> >
> > I would appreciate the author response in which several questions I raised are answered. However, after reading other reviews and responses, I still believe that the issues I pointed out that are answered are not fully addressed (delegated to future works) , so I would like to keep my assessment of 6: marginally above the acceptance threshold

---

### Official Review · Reviewer_atLk · 2022-10-28

**Confidence:** 3
**Clarity, Quality, Novelty And Reproducibility:** Please see the Strengths and Weaknesses.
**Correctness:** 3
**Technical Novelty And Significance:** 2
**Empirical Novelty And Significance:** 2
**Recommendation:** 8

**Strength And Weaknesses:**

Strengths
- This work is well-motivated in terms of how employing supervised graph summarization might help enhance the model in downstream tasks.

- I like how this work presents the limitations of existing work on OT and then attempts to address them with a new method.

- Showing that supervised graph summarization is NP-hard is good information because it encourages researchers to think of fast ways to estimate the solution, which the authors attempted to do.

Weaknesses
- The colors in Table 1 are confusing as colors are usually used to identify which results are state-of-the-art and second place. Also, I found it difficult to associate each result with its corresponding method. I would find it a lot easier to read if there was a column "model" that shows the model for each row of results.
- It is not clear what is meant by OT since most methods are Optimal Transport based. It would be easier to parse the results if a citation is provided next to the method's name such as OT. Is this one by Garg & Jaakkola (2019)?
- Results on larger-scale datasets like ImageNet are missing. Having large scale datasets as part of the benchmark is important since this method's main strength is to compress large graphs while maintaining information that helps in the downstream task.
- It would be useful to know concrete numbers on how many gigabytes of size would be reduced with compression and speed-up in seconds to identify the practical aspect of this work.
- The 100% accuracy on MNIST is a bit suspicious because as far as I know there are few images there that are ambiguous to humans even. Can you elaborate more on how the proposed method got 100% accuracy?
- The results in many cases do not seem significant between the baseline and the proposed method, except on MNIST, can you clarify why that is happening? Also, it seems strange that the compressed graph allows the classifier to train better than the uncompressed version, how did that happen?

Overall this work brings interesting insights and methodology to the table, and I am curious about clarifications on the points I mentioned in the Weaknesses.

**Summary Of The Paper:**

The authors propose a method for summarising graphs in such a way that they are compressed into a smaller representation while retaining the information required for the downstream task. The authors extend an optimum transport-based framework method for graph summarization (Garg & Jaakkola (2019)) to a supervised graph summarization method which also preserves information for the class label by maximising their mutual information. The effectiveness of the proposed method was demonstrated on several datasets such as MNIST outperforming the baseline that is based on optimum transport.

**Summary Of The Review:**

Please see the Strengths and Weaknesses.

---

> ### Author Response · Authors · 2022-11-19
> **Response to Reviewer atLk**
>
> We really appreciate your review on our work and we thank you for your time and careful evaluation of our paper. Please see below our responses to your comments/questions:
>
> **1.**  The colors in Table 1 are confusing
>
> **Response:** We thank the reviewer for the constructive feedback. We have fixed these issues in the revision.
>
> **2.** It is not clear what is meant by OT
>
> **Response:**  Indeed, we are referring to Garg \& Jaakkola (2019), and we will update the citations accordingly, if the style file allows.  However, we are not sure that the reviewer is accurate when they say that ``most methods are Optimal Transport based''.  For instance, looking at the well-cited graph summarization survey Liu et al., 2016, https://arxiv.org/abs/1612.04883, optimal transport methods are not mentioned at all.  Of course, it could be that this is out of date, but a further look via Googling doesn't turn anything up.  Of course, there is extensive work on optimal transport on graphs, but not for graph summarization.
>
> **3.** Results on larger-scale datasets like ImageNet are missing
>
> **Response:** Thank you for the feedback and we have added miniImageNet to our Table 1.
>
> **4.** It would be useful to know concrete numbers
>
> **Response:** The compression ratio is in terms of the number of nodes and is an input to both the (OTC) and our method (so the compression ratios will be the same for both methods when run on the same inputs).  As for the speedup, see Figure1 in our submission which is faster in comparison to the original one that we submitted before. It is because we could have access to a faster system for the revision.
>
> **5.** The 100% accuracy on MNIST is a bit suspicious
>
> **Response:**  You make a fair point, and we should have documented our construction of the MNIST-based dataset precisely (we alluded in the submission to it not being the full dataset, but we explain in full in the revision).  We chose only the first $300$ samples from MNIST in our experiment and as mentioned in the Experimental Study section, all datasets have binary class labels, so the hard instances were likely skipped. To avoid this confusion we have randomly chosen $300$ samples and replaced the experiment in the table. As Table 1 shows 100% accuray is not reported anymore.
>
> **6.** The results in many cases do not seem significant
>
> **Response:**
> 1. Your observation is mostly correct. First, we want to point out that there is a significant difference between the baseline and the proposed method classification errors on the New York taxi data as well.  Furthermore, classification error is not the only important evaluation metric: speed of training and testing are also important.  Our method is uniformly superior to the (OTC) in terms of these metrics, because the baseline performs the optimal transport summarization on each new graph, whereas in our method this is only done on the training set.  However, our original statements about accuracy in the paper were overly strong.  In the revision, we have softened the language and explicitly pointed out the large standard deviations.
>
>  2. As for why the test accuracy is different on MNIST: we hazard a guess that the baseline method underperforms because the graphs involved are grids (so almost all nodes have equal degree), and the initial distribution on nodes in the baseline is based solely on the graph structure (which is not informative).  As for why the same explanation doesn't apply to other image datasets, we hypothesize that this is because our chosen OT parameters only incorporate the local informativeness of each node, and in more complex image classification tasks, where pixels in isolation are less informative than global image structure, this may result in our method failing to perform significantly better than the baseline in terms of test accuracy.
>
> 3. As for how summarization of the graph can lead to better classification performance, this is not so strange, in our view.  Of course, every feature selection or summarization method removes potentially relevant information about the target classification problem, in the information-theoretic sense.  However, the removal of information can make training a specific classifier architecture easier.  If this weren't true, then representation learning wouldn't be a thing (since, by the data processing inequality from information theory, **every** possible transformation of training data either preserves or decreases the mutual information between the classifier input and the class label).
>
> - Please let us know if you have any questions.

---

> > ### Comment · Reviewer_atLk · 2022-12-05
> > **Post Rebuttal**
> >
> > you have addressed most of my concerns, I raise my score to an Accept.

---

### Decision · Program_Chairs · 2023-01-20

**Decision:**

Reject

**Justification For Why Not Higher Score:**

I believe there is lack of knowledge of a substantial amount of older work.

**Justification For Why Not Lower Score:**

The paper presents a reasonable and interesting contribution, but there are some caveats.

**Metareview: Summary, Strengths And Weaknesses:**

This paper presents a creative use of optimal transport for the goal of graph summarization. The reviewers were somewhat divided about this paper, and perhaps the paper needs a bit more work before it is published.

Regarding Theorem 1: I am not sure if the proof of Theorem 1 completely is clear to me. C seems to be sampled once from Bernoulli, and could be either 0 or 1 with probability 1/2 (if I understand correctly the notation C ~ Bernoulli(1/2)), what if it is sampled to be 0, doesn't that mean that p=0 and therefore, there will be no edges at all in G' (both cases three lines above Eq. 4 will lead to no-edge?). This means that overall there is probabiility 1/2 in the reduction that there will be just no edges in G'? Also, what is E_{v,w}, is it $e$ that is referred to a few words before? Also, what is the meaning of the reduction itself being probabilistic in the standard reduction to an NP-hard problem world? Also, what does it mean in the statement "with the knowledge of data distribution?" How can you query the distribution (I assume with an oracle) with this knowledge, and what could be computed? In addition, NP-hardness refers to decision problems, what is the decision problem here you are reducing to? It seems to be a maximization problem. I feel the details and rigor are just a bit fuzzy with this theorem statement and proof. Some of these concerns are echoed in reviewer comments.

All in all, I think the work is in great direction, but given the reviews, discussions with the SAC, and some of the caveats above and others mentioned in the discussion, I feel there needs to be a bit more work done with respect to clarity of notation and logic and related work.

**Summary Of Ac-Reviewer Meeting:**

-